# Colchicine—The Divine Medicine against COVID-19

**DOI:** 10.3390/jpm14070756

**Published:** 2024-07-16

**Authors:** Vanyo Mitev

**Affiliations:** Department of Medical Chemistry and Biochemistry, Medical Faculty, Medical University—Sofia, 1431 Sofia, Bulgaria; vmitev@mu-sofia.bg

**Keywords:** COVID-19, colchicine doses, colchicine toxicity, cytokine storm, NLRP3 inflammasome

## Abstract

Colchicine has a number of effects that suggest it may be useful in the treatment of COVID-19. Myeloid cells are a major source of dysregulated inflammation in COVID-19. The hyperactivation of the NLRP3 inflammasome and the subsequent cytokine storm take place precisely inside them and can lead to multiorgan damage and death. NLRP3 inflammasome inhibition has been assessed at micromolar colchicine concentrations which cannot be achieved in serum. However, colchicine has remarkable ability to accumulate intensively in leukocytes, where the cytokine storm is generated. Over 50 observational studies and randomized clinical trials, small randomized non-controlled trials, and retrospective cohort studies were initiated to test its healing effect in vivo, leading to conflicting, rather disappointing results. The WHO gives a “*Strong recommendation against*” the use of colchicine for COVID-19 treatment. This is because low doses of colchicine are always used, where the concentrations required to inhibit the NLRP3 inflammasome in leukocytes cannot be reached. Considering this, from March 2020, we started the administration of higher doses of colchicine. Our assumption was that a safe increase in colchicine doses to reach micromolar concentrations in leukocytes will result in NLRP3 inflammasome/cytokine storm inhibition. We demonstrated that in 785 inpatients treated with increasing doses of colchicine, mortality fell between two and seven times. Our data, including a large number of COVID-19 outpatients, showed that nearly 100% of the patients treated with this therapeutic regimen escaped hospitalization. In addition, post-COVID-19 symptoms in those treated with colchicine were significantly rarer. As a large number of viruses can overactivate the NLRP3 inflammasome (like seasonal influenza), we are convinced that higher colchicine doses would be useful in these cases as well.

## 1. Introduction

Colchicine is a toxic natural product isolated from the plant *Colchicum autumnale* (autumn crocus or meadow saffron) and *Gloriosa superba* (glory lily), plants of the *Colchiaceae* family. It is a nitrogen-containing substance that is often misdescribed as an alkaloid, although its biosynthetic precursor, demecolcine, is such [1].

Colchicine is one of the earliest drugs described. In an Egyptian medical papyrus of herbal knowledge from 1550 BC (the Ebers Papyrus), it is described as a herbal remedy for joint pain/joint swelling [2,3].

Colchicine’s name may have come from its use as a poison in the district of Colchis of ancient Greece. Medea, the daughter of the king of Colchis and a sorceress, used it as one of her poisons, and it was referred to in Greek mythology as “the destructive fire of the Colchicon Medea” [4,5].

In the first century CE, the Greek physician and pharmacologist Pedanius Dioscorides first described the use of colchicum extract for gout treatment in his pharmacopeia, De Materia Medica [5]. Colchicine has been described in various texts from Persia and Turkey, and it was listed in the London Pharmacopeia from 1618 [5]. Colchicine is the main ingredient of a commercial remedy “Eau Medicinale” created in the 18th century by a French military officer, Nicolas Husson, to treat gout. This product was used successfully by Benjamin Franklin to treat his own gout [4].

Colchicine was first isolated in 1820 by the French chemist Pierre-Joseph Pelletier (1788–1842) and the French chemist and pharmacist Joseph-Bienaimé Caventou (1795–1877) [6]. In 1833, the German chemist and pharmacist Philipp Lorenz Geiger (1785–1836) purified an active ingredient named colchicine [7,8], which has ever since remained in use as a purified natural product. The full synthesis of colchicine was achieved by the Swiss organic chemist Albert Eschenmoser in 1959 [9].

Colchicine was first registered in 1947 in France (“Colchicine capsule”. DailyMed. Retrieved 27 March 2019). Although Benjamin Franklin credited the colchicine introduction to the USA as early as the 18th century [4], it was not until July 2009 that the US Food and Drug Administration approved it as a monotherapy for the treatment of three different indications: familial Mediterranean fever (FMF), acute gout flares, and prophylaxis of gout flares under the Unapproved Drugs Initiative [1,10].

## 2. Other Uses of Colchicine

Colchicine is being used in rheumatology, dermatology, cardiology, and neurology. Beyond being used for treatment of gout and FMF, colchicine is also routinely used in other rheumatic diseases, such as acute flares in calcium pyrophosphate deposition disease (CPPD) [11] and osteoarthritis of the knee [12].

Colchicine is also widely used to treat a variety of dermatological conditions, such as papulosquamous dermatoses, recurrent aphthous stomatitis, Sweet’s syndrome, bullous disease, linear IgA disease, epidermolysis bullosa acquisita, chronic bullous dermatosis of childhood, leucocytoclastic vasculitis, urticarial vasculitis and scleroderma, erythema nodosum leprosum, pyoderma gangrenosum, severe cystic acne, calcinosis cutis, keloids, sarcoid fibromatosis, condyloma acuminata, actinic keratosis, relapsing polychondritis, primary anetoderma, subcorneal pustular dermatosis, erythema nodosum, sclerodema, chronic urticaria, oral aphthosis, genital aphthosis, pseudo-alveolitis, Behçet’s syndrome, and PFAPA syndrome (periodic fever, aphthous stomatitis, pharyngitis, and cervical adenitis). Although colchicine is not a first-line drug for any of the conditions mentioned, it is mostly used early in patients with leukocytoclastic vasculitis, Sweet’s syndrome, and aphthous ulcers [2,12,13,14].

The cardioprotective effects of colchicine are well known and might be due at least in part to its ability to inhibit the pro-thrombotic activity of oxLDL [15] and leukocyte-platelet aggregation [16]. Low colchicine doses are recommended for pericarditis, post-pericardiotomy syndrome, postoperative atrial fibrillation, secondary cardiovascular prevention and in-stent restenosis, stroke prevention, vascular inflammation prevention, myocardial infarction, acute coronary syndrome, coronary artery disease, as well as acute minor ischemic stroke, transient ischemic attack, and prevention of cerebrovascular ischemia in patients with extra- or intracranial atherosclerosis or arteriolosclerosis [12,17,18]. In June 2023, the U.S. FDA approved colchicine (brand name Lodoco), in a dose of 0.5 mg per day that targets residual inflammation, to reduce the risk of further disorders in elderly people with existing atherosclerotic cardiovascular diseases [19].

Colchicine exerts its antiproliferative effects through the inhibition of microtubule formation by blocking the cell cycle at the G2/M phase and triggering apoptosis. Interestingly, comparing cancer incidence in “ever-users” versus “never-users” of colchicine, a significantly lower incidence of all types of cancer (especially prostate and colorectal) was demonstrated in the colchicine “ever-users” when compared to the “never-users” [20]. Also, colchicine has a potential for the palliative treatment of gastric cancer, of hepatocellular carcinoma, and cholangiocarcinoma at high (6 ng/mL), but clinically acceptable colchicine concentration [21]. Recent findings suggest, as well, that colchicine has a therapeutic potential in osteosarcoma [22].

## 3. Why Is Colchicine Being Tested for the Treatment of COVID-19?

Colchicine has a number of effects that suggest it may be useful in the treatment of COVID-19. These effects are synthesized in Figure 1. High colchicine concentrations suppress phagocytosis, PLA2 activation, and lysosomal enzyme release.

In leukocytes, colchicine inhibits microtubule polymerization, reducing their adhesion, recruitment, and activation. This leads to the inhibition of vesicle transport, cytokine secretion, phagocytosis, migration, and division.

In neutrophils, nanomolar concentrations of colchicine alter the E-selectin distribution in vitro, thus reducing cell adhesion, chemotaxis, cell motility, and lysosomal enzyme release during phagocytosis. At a micromolar concentration, it decreases L-selectin expression in vitro and in vivo, which leads to neutrophil–endothelial adhesion inhibition. Colchicine modulates their deformability to suppress neutrophil extravasation and inhibits superoxide anion production and neutrophil infiltration in a dose-dependent manner in vivo. The disruption of microtubule polymerization seems to explain all of these effects. Colchicine increases leukocyte cAMP, which suppresses neutrophil function.

In macrophages, colchicine inhibits the activation of P2X2 and P2X7 receptors, the NLRP3 inflammasome, the release of reactive oxygen species (ROS), and nitrite oxide (NO). It can decrease the number of TNF-α receptors on the surface of macrophages (and endothelial cells). Colchicine activates the nutritional biosensor AMPK, transducing multiple anti-inflammatory effects of both colchicine and RhoA, which suppresses pyrin activity, which in turn inhibits the release of IL-1 and IL-1β, thus reducing the inflammatory process.

Colchicine has been found to significantly inhibit MMP-9, NOX2, and TGF-β1, the levels of which are increased in ST-elevation myocardial infarction patients.

Regarding the effect of colchicine on COVID-19, the most important property is its inhibition of the NLRP3 inflammasome in vitro at micromolar concentrations. It reduces leukocyte inflammation by the inhibition of NLRP3 inflammasome and caspase-1, and, therefore, leads to secondary reductions in cytokines such as 1 L-1β, TNF-α, and IL-6 [14,23,24,25].

## 4. Game of Doses

The “Father of toxicology”, Philippus Aureolus Theophrastus Bombastus von Hohenheim known as Paracelsus (1493–1541) was a German–Swiss *homo universalis*, physician, alchemist, lay theologian, astrologer, and philosopher of the German Renaissance. His interests included medicine, chemistry, and toxicology.

### Sola Dosis Facit Venenum [Paracelsus, Dritte Defensio, 1538]

“Only the dose makes the poison”

Paracelsus was the first to emphasize the importance of dosing in distinguishing between toxicity and treatment [26]. Paracelsus presented the concept of dose response in his Third Defense in response to criticisms against his use of inorganic substances in medicine as too toxic to be used as therapeutic agents [27]. His classic answer was “What is that there is not a poison? All things are poison, and nothing is without poison. Solely the dose determines that a thing is not a poison [18,28,29].

**Variants** (*F*. *Geerk*, *Paracelsus*: *Arzt unserer Zeit* [*Paracelsus*: *Doctor of Our Time*] (*1992*)):“All things are poison, and nothing is without poison; but the dose makes it clear that a thing is not a poison.”“All drugs are poisons; the benefit depends on the dosage.”“All substances are poisons; there is none which is not a poison. The right dose differentiates a poison and a remedy.”

The natural product colchicine is a typical example of Paracelsus’ classic rule that the difference between poison and medicine is the dose. Different doses of colchicine are optimal for different pathological conditions (Figure 2).

## 5. Why Give Higher Doses of Colchicine to Treat Severe COVID-19?

Myeloid cells (granulocytes, monocytes, macrophages, and dendritic cells) are a major source of dysregulated inflammation in COVID-19. The hyperactivation of the NLRP3 inflammasome and the subsequent cytokine storm take place precisely inside them. Expressed in myeloid cells, the NLRP3 inflammasome is a component of the innate immune system and is involved in the activation of many inflammatory processes [34], including the generation of the COVID-19 cytokine storm. NLRP3 inflammasome inhibition has been assessed at colchicine concentrations 10-to 100-fold higher than those achieved in serum [35].

Colchicine has the remarkable ability to accumulate intensively in leukocytes, where the cytokine storm is generated [32,36,37,38]. Colchicine reaches much higher concentrations within leukocytes than in plasma and persists there for several days after ingestion [38]. Whereas the peak plasma concentration after a single oral dosing of 0.6 colchicine is approximately 3 nmol/L, it has been shown to accumulate in a saturable manner in neutrophils 40 to 200 nmol/L [39]. One possible explanation for this phenomenon in the neutrophils is their low expression of the *ABCB1* (ATP Binding Cassette Subfamily B Member 1) drug transporter gene [36]. Colchicine concentrations in neutrophils are three-fold higher than those in lymphocytes, probably due to the reduced expression of P-glycoprotein (P-gp), whose function is to remove colchicine from cells [40].

Thus, it is logical to expect that by raising the doses of colchicine within acceptable limits, a level of its concentration sufficient to inhibit the NLRP3 inflammasome in leukocytes can be reached. Increasing doses of colchicine can lead to such an accumulation in macrophages, neutrophils, and monocytes that is sufficient to inhibit the NLRP3 inflammasome and, accordingly, the cytokine storm (Figure 3).

## 6. In Search of the Optimal Dose of Colchicine to Inhibit the NLRP3 Inflammasome

### Colchicine Toxicity

Colchicine is well known for its narrow therapeutic index and no clear-cut line between non-toxic, toxic, and lethal doses. Finkelstein et al. conducted a large-scale study spanning 44 years in search of the minimal lethal dose of colchicine and concluded that ‘the lowest reported lethal doses of oral colchicine have ranged from 7 to 26 mg’ [41]. This opinion has been accepted as dogma and is being quoted continuously [42].

However, our analysis showed that the deaths reported in the literature with colchicine doses of 7–7.5 mg are due to drug interactions and to a large extent, this also applies to the described lethal doses of colchicine of 15–18 mg. We proposed that the classical sentence of Finkelstein et al., 2010, should be recast as ‘The lowest reported lethal doses of oral colchicine has ranged from 15 to 26 mg’ [42]. We consider that doses of colchicine below 0.1 mg/kg are completely safe, and those between 0.1 and 0.2 mg/kg may lead to certain toxicity side effects in some cases, but not to death [42].

More recent examples can be cited as a confirmation of this. In the clinical reports published between 2001 and 2021 and assessing colchicine application and toxicity in adults, no cases can be found refuting our suggestion of a lowest lethal dose of colchicine [43]. The 56-year-old man who died after taking 12 mg (0.17 mg/kg) of colchicine had a medical history of gout and poor kidney function due to chronic kidney disease. Furthermore, he was late-diagnosed and admitted to hospital after drinking alcohol [44]. In another deadly case, a 33-year-old woman was admitted after ingesting 20 mg (0.33 mg/kg) of colchicine combined with atorvastatin, ibuprofen, diclofenac, and furosemide [45]. Statins are inhibitors of the cytochrome P-450 (CYP) 3A4 isozyme and P-glycoprotein, so that, taken together with colchicine, they represent a clinically significant interaction.

A number of cases of colchicine overdose in the range of 15–24 mg with complete recovery have been described: an 18-year-old female had ingested 18 mg (~0.4 mg/kg) of colchicine in a suicide attempt [46]; a 48-year-old Australian Caucasian male had ingested more than >10 mg colchicine [47]; a 51-year-old male with medical history of depression, high blood pressure, and gout treated with colchicine had ingested 17 mg of *Colchimax* [48]; a 15-year-old girl survived after the ingestion of 24 mg of colchicine [49]; a 38-year-old male had ingested 15 mg (0.2 mg/kg) [50]. All these patients showed varying degrees of toxicity, and our opinion that, within these limits, deaths occur only if drug interactions or renal and/or hepatic damage are present, is confirmed [42]. Two recent retrospective studies confirm our observations [51,52]. In a series of 21 cases of colchicine poisoning, a retrospective cross-sectional study demonstrated that the lowest lethal dose was 20 mg in an accidental intake in a 10-year-old boy with G6PD deficiency. The other two non-survivors had ingested 38 and 40 mg of colchicine [51].

Stamp et al., 2023 demonstrated 48 cases of colchicine poisoning, some of which are commented above and in [52]. Death rarely occurs at doses up to 25 mg of colchicine unless other aggravating factors are present, most commonly drug interactions and/or kidney or liver damage. A typical case is the death of a 39-year-old female with bipolar affective disorder, hypothyroidism, reflux disease, multiple suicide attempts involving multidrug overdose, and illicit drug use who ingested 25 mg (0.28 mg/kg) of colchicine together with 50 tablets of indomethacin (1.25 g) and 10 tablets of zopiclone (75 mg) as well as topiramate, valproic acid, olanzapine, lorazepam, levothyroxine, oxycodone, furosemide, and omeprazole (unknown timing and doses) prior to admission [53].

Colchicine is a safe and well-tolerated drug which increases the rate of diarrhea and gastrointestinal adverse events but does not increase the rate of liver, sensory, muscle, infectious, or hematological adverse events or death [54].

## 7. Our Experience in Treating COVID-19 with a High Dose of Colchicine

In March 2020, we started the administration of higher doses of colchicine due to the fact that we did not get satisfactory results with low doses of the drug. Our assumption was that a safe increase in colchicine doses to reach micromolar concentrations in leukocytes will result in NLRP3 inflammasome inhibition.

The inhibitory effect of colchicine on the cytokine storm occurs at doses such as [0.5 mg per 10 kg body weight] − 0.5 mg, but not at more than a 5 mg loading dose. This results in 0.04–0.045 mg/colchicine/kg [55,56,57]. We consider that doses of colchicine below 0.1 mg/kg are completely safe, and those between 0.1 and 0.2 mg/kg may lead to toxicity side effects in some cases, but not to death [42].

High doses of colchicine were administered in four hospitals on different schedules. A significant reduction in mortality was observed everywhere, with the higher doses of colchicine being lower [56,57].

## 8. Colchicine Therapeutic Regimens

### 8.1. Hospital I

The therapeutic dose of colchicine is calculated according to the formula [0.5 mg per 10 kg body weight] − 0.5 mg, but not at more than a 5 mg loading dose. This results in 0.04–0.045 mg/colchicine/kg. The maintenance dose (up to day 15) is half the loading dose. Mortality is reduced 4.8-fold [56].

### 8.2. Hospital II

Colchicine: day 1—3.5–4 mg of colchicine; day 2—3 mg; day 3—2.5 mg; days 4–15—1.5 mg. Mortality is reduced 4.3-fold [56].

### 8.3. Hospital III

Colchicine: 4X1 (2 mg)—4 days; 3X1(1.5 mg)—4 days; 2X1(1 mg) up to day 15. Mortality is reduced 2.3-fold [58].

Colchicine 4X2 (4 mg)—4 days; 3X2(3 mg)—4 days; 2X2(2 mg)—4 days; 2X1(1 mg) up to day 15. Mortality is reduced 4-fold [57].

### 8.4. Hospital IV

Colchicine: day 1—4 mg; day 2—3.5 mg; day 3—3 mg; day 4—2.5 mg; day 5—2 mg; day 6—1.5 mg; day 7—1 mg, up to day 30. Mortality is reduced 6.7-fold [57].

## 9. Case Series of Severe or Critical COVID-19 Dramatically Affected by High Dose of Colchicine

We have published a series of cases demonstrating the life-saving effect of colchicine. For example, a patient in whom the chest computed tomography demonstrated that only about 10% of the lung was not affected by severe bilateral pneumonia and acute respiratory distress syndrome recovered after treatment with a high dose of colchicine [59].

Obese patients with a BMI of over 40 represent perhaps the most at-risk group for COVID-19 complications. A 120 kg patient with type 2 diabetes mellitus, hypertension, and gout was hospitalized on the third day of the COVID-19 diagnosis in relatively good general condition (oxygen saturation was 89%). Despite the started standard therapy, the patient continued to deteriorate (oxygen saturation was 74%) and on the eighth day, colchicine was included at a dose of 6 mg. The patient quickly recovered [60]. Three high-risk patients with a BMI of over 50 and 60 also responded dramatically to 5 mg of colchicine [61]. Very demonstrative is the case of a 101-year-old 70-kg man fighting for his life after two surgical interventions. While recovering in the intensive care unit, he was infected with COVID-19, but thanks to immediate treatment with 4 mg of colchicine (0.06 mg/kg), the patient was saved [58]. These are typical cases of the life-saving effect of high doses of colchicine in high-risk COVID-19 patients.

It is interesting to note that an accidentally taken single overdose of colchicine (15 mg or 12.5 mg) is sufficient for the complete recovery of patients, including the eradication of pericardial effusion [62].

## 10. In 785 Inpatients Treated with Increasing Doses of Colchicine, Mortality Fell between Two and Seven Times

In the already-described 452 inpatients, higher colchicine doses reduced the mortality about five-fold [56]. In another 333 inpatients treated with different, but high colchicine doses, a clear reduction in the mortality between two- and seven-fold has emerged with increasing doses of colchicine. Thus, colchicine loading doses of 4 mg are more effective than those of 2 mg [57]. Despite the higher than the so-called “standard doses” of colchicine, our doses are completely safe [42].

Our data, including also a large number of COVID-19 outpatients, showed that nearly 100% of the patients treated with this therapeutic regimen escaped hospitalization. In addition, post-covid symptoms in those treated with colchicine were significantly rarer (Figure 4). These results have been announced in several national and international conferences and are being processed for publication [24].

## 11. Discussion

### 11.1. How to Prevent the Development of Severe COVID-19?

The World Health Organization (WHO) recommends the antiviral drugs Paxlovid, Molnupiravir, and Remdesivir. The low-potency and mutagenically questionable Molnupinavir (Lagevrio) from Merck&Co Inc. was rejected by the European Medicines Agency (EMA) (https://www.ema.europa.eu/en/medicines/human/withdrawnapplications/lagevrio, accessed on 10 June 2024).

Remdesivir may cause serious allergic reactions, including infusion-related reactions and anaphylaxis, drug-induced liver injury leading to acute liver failure, and death [63]. According to the instructions of Mayo Clinic (Rochester, MN, USA), “this medicine is to be given only by or under the immediate supervision of your doctor” (https://www.mayoclinic.org/drugs-supplements/remdesivir-intravenous-route/side-effects/drg-20503608, accessed on 10 June 2024).

### 11.2. The Rise and Fall of Paxlovid

The WHO’s favorite remains to be ritonavir-boosted nirmatrelvir (Paxlovid), which is “*strongly recommended in favor*” [24].

However, this miracle drug against COVID-19 has recently suffered a real meltdown. The reason for this is the EPIC-SR, which is “a double-blind placebo controlled RCT that shows no benefit from Paxlovid in a vaccinated population, and in unvaccinated patients without risk factors”. “The time to sustained alleviation of all signs and symptoms of COVID-19 did not differ significantly between participants who received nirmatrelvir–ritonavir and those who received placebo” [64]. Pfizer knew that Paxlovid did not work in 2022 but waited until 4 April, 2024 to publish this study. According to Dr Morgenstern: “We have known this trial was negative for 2 years, despite Pfizer trying to hide the data from us”, and he continued, “This data is awful. I don’t think we would have started using Paxlovid if we weren’t in the middle of a pandemic” [65]. Very importantly, despite being a “standard risk” trial, the EPIC-SR authors state that 50% of this group were actually high risk!

In “The Rise and Fall of Paxlovid”, Dr. Paul Sax “recap the astonishing success and now failure” of this drug [66].

At the end of 2021, the EPIC-HR study of high-risk outpatients with COVID-19 demonstrated that Paxlovid-treated participants had an 88.9% reduction in risk of hospitalization or death [67]. It turned out that these percentages varied widely from 88.9% to 26% [68]. In addition, Paxlovid was not effective in hospitalized patients [69], did not reduce the risk of developing long COVID [70], and caused rebounds, or just did not prevent them [66].

Now, the guidelines recommend Paxlovid only for persons who are at high risk for disease progression, but how to explain that it is not effective in those 50% of the EPIC-SR trial, who were actually in the high-risk group?

The failure of antivirals to solve the problem of preventing COVID-19 complications is due to the fact that there is no direct link between viral replication and the hyperresponsiveness of the NLRP3 inflammasome [24].

### 11.3. How the Opportunity to Save Millions of Human Lives Was Missed Because One Point Instead of a Comma Was Used. “The Fall and Rise of Colchicine”

Given the multiple anti-inflammatory and anti-COVID-19 effects of colchicine in vitro, over 50 observational studies and randomized clinical trials, small randomized non-controlled trials, and retrospective cohort studies were initiated to test its healing effect in vivo, leading to conflicting results [24,71,72].

The Randomized Evaluation of COVID-19 thERapY (RECOVERY) trial, launched on 23 March, 2020, is a large-scale, randomized, controlled trial, which claims to be the most credible study on the effect of various proposed drugs to treat COVID-19. The RECOVERY Trial is presented as “an exceptional study that is leading the global fight against COVID-19”, that “has saved hundreds of thousands—if not millions—of lives worldwide”. It encompasses 48941 participants and 187 active sites [73]. The WHO automatically complies with the conclusions of these clinical trials and gives “*Strong recommendation against*” the use of colchicine for COVID-19 treatment (https://www.who.int/publications-detail-redirect/WHO-2019-nCoV-therapeutics-2022.4/, accessed on 10 June 2024).

The analysis based on 2178 reported deaths among 11,162 randomized patients concluded that “To date there has been no convincing evidence of the effect of colchicine on clinical outcomes in patients admitted to hospital with COVID-19” [74]. It is very important to note that only low doses of colchicine were used.

The co-Chief Investigator, Sir Peter Horby, said: ‘This is the largest ever trial of colchicine… Whilst we are disappointed that the overall result is negative…’ [73]. This is misleading. The following is correct: ‘This is the largest ever trial of colchicine with low doses…’

The other co-Chief Investigator for the RECOVERY Trial, Sir Martin Landray, said: ‘So, it is disappointing that colchicine, which is widely used to treat gout and other inflammatory conditions, has no effect in these patients’ [73].

This conclusion is misleading. The correct conclusion should sound like this: ‘So, it is disappointing that colchicine, which is widely used to treat gout and other inflammatory conditions, has no effect in these patients, at low doses’. Here is how one “point” instead of “comma” doomed the colchicine treatment of COVID-19 to failure.

However, high but safe doses of colchicine solve the problem of ambulatory treatment of COVID-19 and reduce the mortality of inpatients many times over [22,42,55,56,57,58,59,60,61]. We are convinced that the strategy of Big Pharma and the WHO to treat COVID-19 is wrong. They put emphasis on antiviral agents through blocking the effect of various cytokines. We believe that to avoid the CS, inhibition of the NLRP3 inflammasome is mandatory. Once this happens, the cytokine level will drop or not rise at all. This renders the use of expensive antibodies that inhibit the effect of important cytokines such as IL-6 (tocilizumab) pointless. Colchicine does this, but at high doses. Big Pharma should focus on drugs that block the NLRP3 inflammasome, as well as on those that inhibit the virus entering the cell. The latter was being done excellently so far by bromhexine hydrochloride, when taken prophylactically for at least a month, during the wave of COVID-19 [24,57].

A large number of viruses can overactivate the NLRP3 inflammasome [75]. For exemple, according to the World Health Organization (WHO), the seasonal influenza causes 3–5 million cases of severe illnesses and between 290,000 and 650,000 deaths per year globally, with 36% of these deaths taking place in low- and middle-income countries (Influenza (Seasonal). World Health Organization. 3 October 2023 (https://www.who.int, accessed on 10 June 2024).

We are convinced that higher colchicine doses would be useful in these cases as well [60]. We set out to test this hypothesis with success.

## 12. Conclusions

A number of questions remain without answers. The RECOVERY Trial co-Chief Investigators, the Lancet reviewers, and the Lancet editors should answer the following questions: Why is Paracelsus’s rule not being followed and why do attempts to prove an effect at low doses of colchicine continue to this day? Why were only colchicine doses recommended for gout given? COVID-19 is not gout! Why has the study of Terkeltaub et al., 2010, with low and high doses of colchicine not been repeated? [32]. Why do attempts to disprove Einstein that “Insanity is doing the same thing over and over and expecting different results” continue? Which is preferable—alive with diarrhea for a few days or dead without diarrhea?

Thus, the opportunity to save millions of human lives was missed.

## Figures and Tables

**Figure 1 jpm-14-00756-f001:**
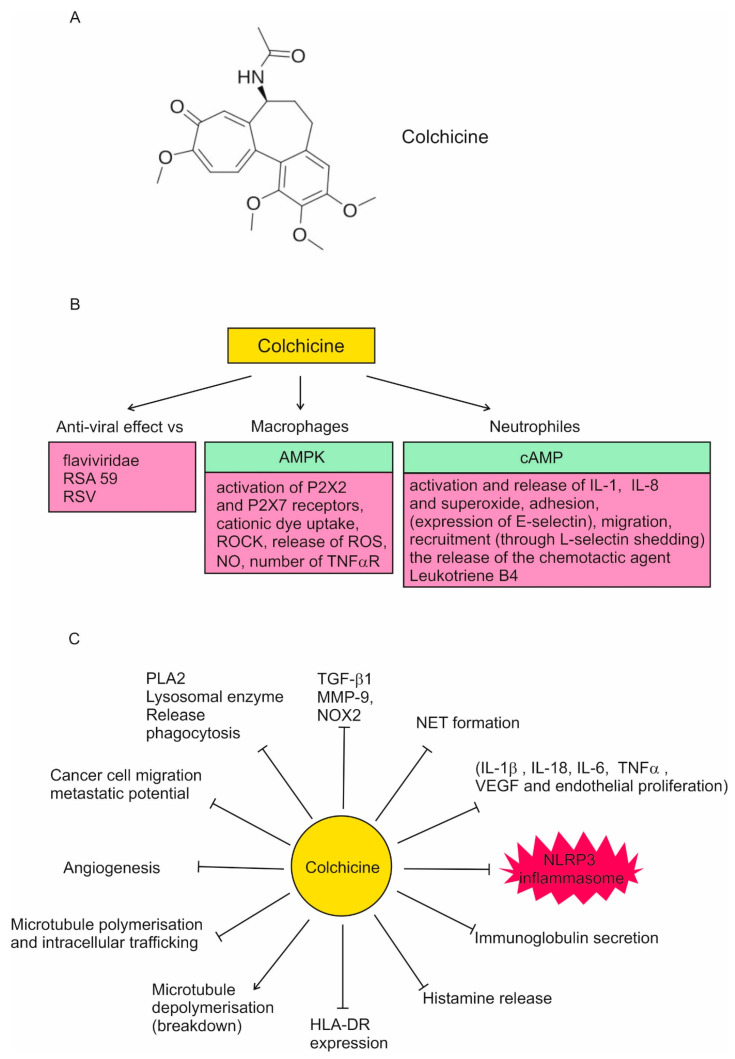
Colchicine effects. (**A**) Colchicine has one stereocenter located at carbon 7 and its natural configuration is S. The molecule also contains one chiral axis, whose natural configuration is aS. Colchicine has four stereoisomers, but the only one found in nature is the aS, 7s configuration. Colchicine accumulates in white blood cells, decreasing their motility, mobilization (especially chemotaxis), adhesion, and, very important in the case of COVID-19 pathophysiology, it inhibits the NLRP3 inflammasome. (**B**) The stimulating effects of colchicine. (**C**) The inhibitory effects of colchicine. Green color—stimulation; red color—inhibition; P2X2, P2X7—Purinergic Receptors; ROCK—*Rho*-associated protein *kinase*; ROS—Reactive oxygen species; NO—Nitric Oxide; PLA2—Phospholipase A2; NOX2—NADPH oxidase 2; RSA59—Isogenic recombinant demyelinating strain of mouse hepatitis virus (MHV); RSV—Respiratory Syncytial Virus; MMP9—Matrix metalloproteinase-9; NET—Neutrophil Extracellular Traps; TGF-β1—transforming growth factor beta 1; VEGF—vascular endothelial growth factor; TNFa—tumor necrosis factor alpha. For references see [14,23,24,25].

**Figure 2 jpm-14-00756-f002:**
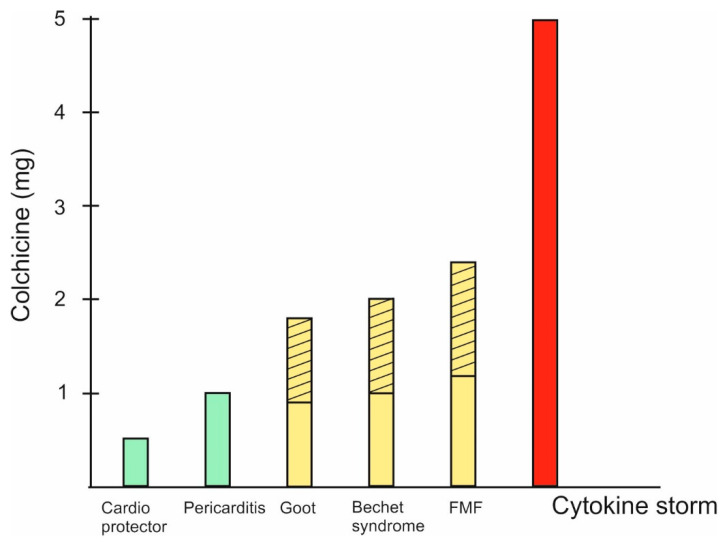
Doses of colchicine in different pathological situations. 1. For cardio protection, colchicine doses of 0.5/0.6 mg daily for 6 months are recommended [19]; for stroke prevention, 0.5–1 mg once daily [30]; and for vascular inflammation prevention, 0.5 mg once daily for 60 months [31]. 2. For ccute coronary syndrome, coronary artery disease, pericarditis, or atrial fibrillation, a dose of 0.5 mg twice daily for 1 month to 1 year or 0.5 mg once daily for a median of 3 years is recommended [31]. 3. For Behçet’s syndrome, a dose of 1–2 mg daily for 3 months is recommended [31]. 4. For acute gout, a dose of 1.8 mg total is recommended [32]. 5. The highest recommended doses for FMF are up to 2.4 mg. Interestingly, in cases where there is no effect, they are recommended “the maximum tolerated dose” [33]. 6. For COVID-19, a loading dose of up to 5 mg is recommended [24]. (Green bars represent low doses, yellow—medium, red—high. Diagonal stripes show varying therapeutic dosage).

**Figure 3 jpm-14-00756-f003:**
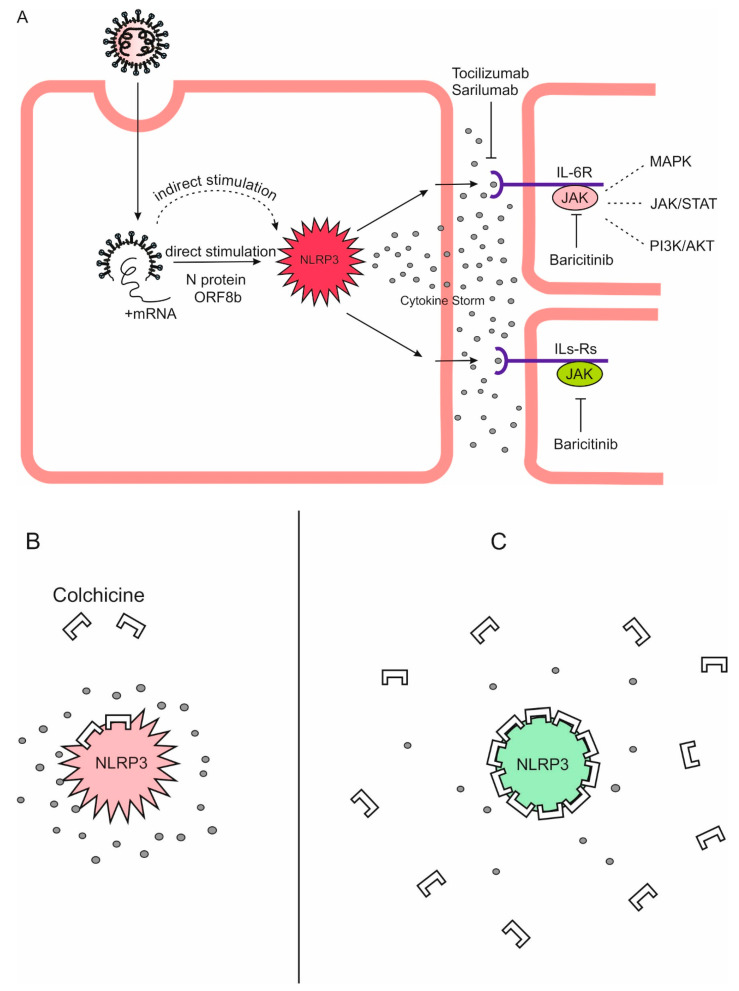
The theoretical basis of high-dose colchicine treatment in COVID-19. Mortality in COVID-19 is due to a cytokine storm triggered directly and indirectly by SARS-CoV-2, which hyperactivates the NLRP3 inflammasome in myeloid cells. It is inhibited by micromolar concentrations of colchicine. As colchicine has a remarkable ability to accumulate intensively in leukocytes, its increasing doses can lead to such an accumulation in macrophages, neutrophils, and monocytes that is sufficient to inhibit the NLRP3 inflammasome and, accordingly, the cytokine storm. (**A**) Direct and indirect stimulation of NLRP3 inflammasome by SARS-CoV-2 can lead to its hyperactivation, cytokine storm, multiorgan failure, and death. (**B**) Low doses of colchicine are not sufficient for NLRP3 inflammasome/cytokine storm inhibition. (**C**) High doses of colchicine are capable of inhibiting the NLRP3 inflammasome, interrupting the cytokine storm. (Red NLRP3—hyperactivation; green NLRP3—normal function).

**Figure 4 jpm-14-00756-f004:**
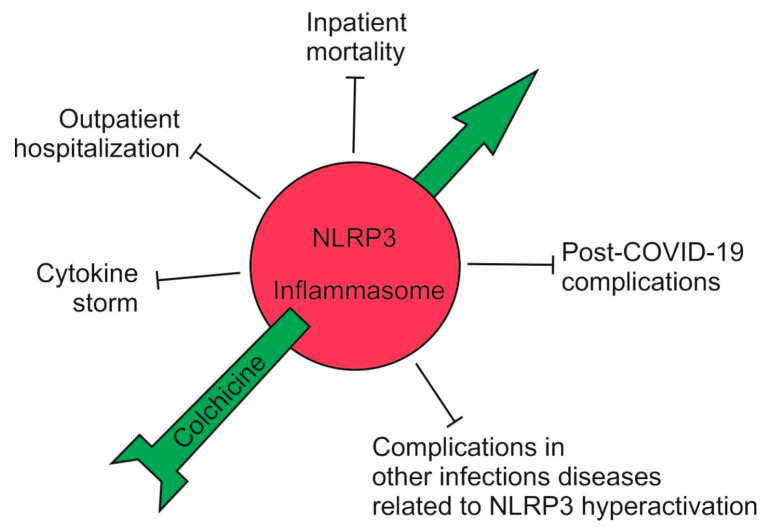
The effects of high-dose colchicine on the course of COVID-19. A high dose of colchicine inhibited the NLRP3 inflammasome and, accordingly, the cytokine storm; outpatients did not develop complications and avoided hospitalization; the mortality of hospitalized patients decreased up to seven-fold; and post-COVID-19 symptoms decreased sharply. A high dose of colchicine should also be effective in other infectious conditions associated with hyperactivation of the NLRP3 inflammasome (Red—hyperactivation; green—normal function).

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
