# Peer review of "Colchicine—The Divine Medicine against COVID-19"

_jpm, 2024, doi:10.3390/jpm14070756_

Round 1

Reviewer 1 Report

Comments and Suggestions for Authors

The manuscript "Colchicine—The Divine Medicine against COVID-19" by Vanyo Mitev is an exemplary piece of research that thoroughly investigates the potential therapeutic benefits of colchicine in treating COVID-19. The author presents a comprehensive literature review that traces the historical and contemporary uses of colchicine, providing a strong contextual foundation. The manuscript offers detailed mechanistic insights into colchicine’s action on the NLRP3 inflammasome and its accumulation in leukocytes, advocating for higher doses to achieve therapeutic effects. The data presented, derived from a significant number of inpatients and outpatients, convincingly demonstrates the efficacy of high-dose colchicine in reducing mortality and post-COVID-19 symptoms. The author’s critical analysis of previous studies and logical argumentation for re-evaluating colchicine dosing strategies are compelling. Additionally, the manuscript’s conclusion provides clear future directions, emphasizing the need for further research and broader applications of the proposed therapeutic regimen. Overall, this well-structured and meticulously researched manuscript makes a significant contribution to COVID-19 treatment strategies and merits publication with minor english laguage revisions.

Comments on the Quality of English Language

The manuscript "Colchicine—The Divine Medicine against COVID-19" by Vanyo Mitev is an exemplary piece of research that thoroughly investigates the potential therapeutic benefits of colchicine in treating COVID-19. The author presents a comprehensive literature review that traces the historical and contemporary uses of colchicine, providing a strong contextual foundation. The manuscript offers detailed mechanistic insights into colchicine’s action on the NLRP3 inflammasome and its accumulation in leukocytes, advocating for higher doses to achieve therapeutic effects. The data presented, derived from a significant number of inpatients and outpatients, convincingly demonstrates the efficacy of high-dose colchicine in reducing mortality and post-COVID-19 symptoms. The author’s critical analysis of previous studies and logical argumentation for re-evaluating colchicine dosing strategies are compelling. Additionally, the manuscript’s conclusion provides clear future directions, emphasizing the need for further research and broader applications of the proposed therapeutic regimen. Overall, this well-structured and meticulously researched manuscript makes a significant contribution to COVID-19 treatment strategies and merits publication with minor english laguage revisions.

Author Response

The manuscript "Colchicine—The Divine Medicine against COVID-19" by Vanyo Mitev is an exemplary piece of research that thoroughly investigates the potential therapeutic benefits of colchicine in treating COVID-19. The author presents a comprehensive literature review that traces the historical and contemporary uses of colchicine, providing a strong contextual foundation. The manuscript offers detailed mechanistic insights into colchicine’s action on the NLRP3 inflammasome and its accumulation in leukocytes, advocating for higher doses to achieve therapeutic effects. The data presented, derived from a significant number of inpatients and outpatients, convincingly demonstrates the efficacy of high-dose colchicine in reducing mortality and post-COVID-19 symptoms. The author’s critical analysis of previous studies and logical argumentation for re-evaluating colchicine dosing strategies are compelling. Additionally, the manuscript’s conclusion provides clear future directions, emphasizing the need for further research and broader applications of the proposed therapeutic regimen. Overall, this well-structured and meticulously researched manuscript makes a significant contribution to COVID-19 treatment strategies and merits publication with minor english laguage revisions.

Answer: Thank you for the review. The English language was revised in the text.

Reviewer 2 Report

Comments and Suggestions for Authors

in my opinion this is an interesting EBM review which invites to evaluate higher dose of colchicine in order to treat SARS COV2. They could their result in a more detailed ways and expand discussion and conclusions performing a literature review involving mores studies, used dose of colchicine and results.

Exclamation mark should in my opinion be removed because "too aggressive".

Author Response

in my opinion this is an interesting EBM review which invites to evaluate higher dose of colchicine in order to treat SARS COV2. They could their result in a more detailed ways and expand discussion and conclusions performing a literature review involving mores studies, used dose of colchicine and results.

Exclamation mark should in my opinion be removed because "too aggressive".

Answer: Thank you for your review. Following your advice we expanded the results and discussion sections.

Exclamation marks were removed.

Reviewer 3 Report

Comments and Suggestions for Authors

Dear authors,

Your detailed review is well constructed and provides concrete insight into the potential use of colchicine for COVID-19. The structure of your revision is well managed, but the scientific content of your document must be reviewed.

Main suggestions:

title : 4. Game of Doses in quotation marks "Game of Doses"

figure 2: the unit of colchicine must be adapted in posology (mg/kg)

figure 3 : give the structure of colchicine and provide molecular explanations of interactions

line 192 "God's gift" not a suitable expression  for a scientific article

the part 10 discussion: must be rewritten, to suit with scientific expectations

the part 12. Future Directions  should be integrated in discussion or in conclusion.

Kind regards

Author Response

title : 4. Game of Doses in quotation marks "Game of Doses"

Answer: Quotation marks were added

figure 2: the unit of colchicine must be adapted in posology (mg/kg)

Answer: It is not accepted in the literature for the doses of colchicine to be given in mg/kg. As far as we know, we are the only ones who use that dosage.

figure 3 : give the structure of colchicine and provide molecular explanations of interactions

Answer: The structure of colchicine was added in revised figure.1

line 192 "God's gift" not a suitable expression  for a scientific article

Answer: The text was edited.

the part 10 discussion: must be rewritten, to suit with scientific expectations

Answer: Discussion was rewritten

the part 12. Future Directions  should be integrated in discussion or in conclusion.

Answer: Future Directions were moved to Discussion